# RG-FLOW: A HIERARCHICAL AND EXPLAINABLE FLOW MODEL BASED ON RENORMALIZATION GROUP AND SPARSE PRIOR

## ABSTRACT

Flow-based generative models have become an important class of unsupervised learning approaches. In this work, we incorporate the key idea of renormalization group (RG) and sparse prior distribution to design a hierarchical flow-based generative model, called RG-Flow, which can separate information at different scales of images with disentangled representations at each scale. We demonstrate our method mainly on the CelebA dataset and show that the disentangled representations at different scales enable semantic manipulation and style mixing of the images. To visualize the latent representations, we introduce receptive fields for flow-based models and find that the receptive fields learned by RG-Flow are similar to those in convolutional neural networks. In addition, we replace the widely adopted Gaussian prior distribution by a sparse prior distribution to further enhance the disentanglement of representations. From a theoretical perspective, the proposed method has $O(\log L)$ complexity for image inpainting compared to previous generative models with $O(L^2)$ complexity.

## 1 INTRODUCTION

One of the most important unsupervised learning tasks is to learn the data distribution and build generative models. Over the past few years, various types of generative models have been proposed. Flow-based generative models are a particular family of generative models with tractable distributions (Dinh et al., 2017; Kingma & Dhariwal, 2018; Chen et al., 2018b; 2019; Behrmann et al., 2019; Hoogeboom et al., 2019; Brehmer & Cranmer, 2020; Rezende et al., 2020; Karami et al., 2019). Yet the latent variables are on equal footing and mixed globally. Here, we propose a new flow-based model, RG-Flow, which is inspired by the idea of *renormalization group* in statistical physics. RG-Flow imposes locality and hierarchical structure in bijective transformations. It allows us to access information at different scales in original images by latent variables at different locations, which offers better explainability. Combined with sparse priors (Olshausen & Field, 1996; 1997; Hyvärinen & Oja, 2000), we show that RG-Flow achieves hierarchical disentangled representations.

Renormalization group (RG) is a powerful tool to analyze statistical mechanics models and quantum field theories in physics (Kadanoff, 1966; Wilson, 1971). It progressively extracts more coarse-scale statistical features of the physical system and decimates irrelevant fine-grained statistics at each scale. Typically, the local transformations used in RG are designed by human physicists and they are not bijective. On the other hand, the flow-based models use cascaded invertible global transformations to progressively turn a complicated data distribution into Gaussian distribution. Here, we would like to combine the key ideas from RG and flow-based models. The proposed RG-flow enables the machine to learn the optimal RG transformation from data, by constructing local invertible transformations and build a hierarchical generative model for the data distribution. Latent representations are introduced at different scales, which capture the statistical features at the corresponding scales. Together, the latent representations of all scales can be jointly inverted to generate the data. This method was recently proposed in the physics community as NeuralRG (Li & Wang, 2018; Hu et al., 2020).

Our main contributions are two-fold: First, RG-Flow can separate the signal statistics of different scales in the input distribution naturally, and represent information at each scale in its latent vari-

ables $z$. Those hierarchical latent variables live on a hyperbolic tree. Taking CelebA dataset (Liu et al., 2015) as an example, the network will not only find high-level representations, such as the gender factor and the emotion factor for human faces, but also mid-level and low-level representations. To visualize representations of different scales, we adopt the concept of *receptive field* from convolutional neural networks (CNN) (LeCun, 1988; LeCun et al., 1989) and visualize the hidden structures in RG-flow. In addition, since the statistics are separated into a hierarchical fashion, we show that the representations can be mixed at different scales. This achieves an effect similar to style mixing. Second, we introduce the *sparse prior distribution* for latent variables. We find the sparse prior distribution is helpful to further disentangle representations and make them more explainable. The widely adopted Gaussian prior is rotationally symmetric. As a result, each of the latent variables in a flow model usually does not have a clear semantic meaning. By using a sparse prior, we demonstrate the clear semantic meaning in the latent space.

## 2 RELATED WORK

Some flow-based generative models also possess multi-scale latent space (Dinh et al., 2017; Kingma & Dhariwal, 2018), and recently hierarchies of features have been utilized in Schirrmeister et al. (2020), where the top-level feature is shown to perform strongly in out-of-distribution (OOD) detection task. Yet, previous models do not impose hard locality constraint in the multi-scale structure. In Appendix C, the differences between globally connected multi-scale flows and RG-Flow are discussed, and we see that semantic, meaningful receptive fields do not show up in the globally connected cases. Recently, other more expressive bijective maps have been developed (Hoogeboom et al., 2019; Karami et al., 2019; Durkan et al., 2019), and those methods can be incorporated into the proposed structure to further improve the expressiveness of RG-Flow.

Some other classes of generative models rely on a separate inference model to obtain the latent representation. Examples include variational autoencoders (Kingma & Welling, 2014), adversarial autoencoders (Makhzani et al., 2015), InfoGAN (Chen et al., 2016), and BiGAN (Donahue et al., 2017; Dumoulin et al., 2017). Those techniques typically do not use hierarchical latent variables, and the inference of latent variables is approximate. Notably, recent advances suggest that having hierarchical latent variables may be beneficial (Vahdat & Kautz, 2020). In addition, the coarse-to-fine fashion of the generation process has also been discussed in other generative models, such as Laplacian pyramid of adversarial networks (Denton et al., 2015), and multi-scale autoregressive models (Reed et al., 2017).

*Disentangled representations* (Tenenbaum & Freeman, 2000; DiCarlo & Cox, 2007; Bengio et al., 2013) is another important aspect in understanding how a model generates images (Higgins et al., 2018). Especially, disentangled high-level representations have been discussed and improved from information theoretical principles (Cheung et al., 2015; Chen et al., 2016; 2018a; Higgins et al., 2017; Kipf et al., 2020; Kim & Mnih, 2018; Locatello et al., 2019; Ramesh et al., 2018). Apart from the high-level representations, the multi-scale structure also lies in the distribution of natural images. If a model can separate information of different scales, then its multi-scale representations can be used to perform other tasks, such as style transfer (Gatys et al., 2016; Zhu et al., 2017), face mixing (Karras et al., 2019; Gambardella et al., 2019; Karras et al., 2020), and texture synthesis (Bergmann et al., 2017; Jetchev et al., 2016; Gatys et al., 2015; Johnson et al., 2016; Ulyanov et al., 2016).

Typically, in flow-based generative models, Gaussian distribution is used as the prior for the latent space. Due to the rotational symmetry of Gaussian prior, an arbitrary rotation of the latent space would lead to the same likelihood. Sparse priors (Olshausen & Field, 1996; 1997; Hyvärinen & Oja, 2000) was proposed as an important tool for unsupervised learning and it leads to better explainability in various domains (Ainsworth et al., 2018; Arora et al., 2018; Zhang et al., 2019). To break the symmetry of Gaussian prior and further improve the explainability, we introduce a sparse prior to flow-based models. Please refer to Figure 12 for a quick illustration on the difference between Gaussian prior and the sparse prior, where the sparse prior leads to better disentanglement.

*Renormalization group* (RG) has a broad impact ranging from particle physics to statistical physics. Apart from the analytical studies in field theories (Wilson, 1971; Fisher, 1998; Stanley, 1999), RG has also been useful in numerically simulating quantum states. The multi-scale entanglement renormalization ansatz (MERA) (Vidal, 2008; Evenbly & Vidal, 2014) implements the hierarchical structure of RG in tensor networks to represent quantum states. The exact holographic mapping (EHM)

(Qi, 2013; Lee & Qi, 2016; You et al., 2016) further extends MERA to a bijective (unitary) flow between latent product states and visible entangled states. Recently, Li & Wang (2018); Hu et al. (2020) incorporates the MERA structure and deep neural networks to design a flow-base generative model that allows machine to learn the EHM from statistical physics and quantum field theory actions. In quantum machine learning, recent development of quantum convolutional neural networks also (Cong et al., 2019) utilize the MERA structure. The similarity between RG and deep learning has been discussed in several works (Bény, 2013; Mehta & Schwab, 2014; Bény & Osborne, 2015; Oprisa & Toth, 2017; Lin et al., 2017; Gan & Shu, 2017). The information theoretic objective that guides machine-learning RG transforms are proposed in recent works (Koch-Janusz & Ringel, 2018; Hu et al., 2020; Lenggenhager et al., 2020). The meaning of the emergent latent space has been related to quantum gravity (Swingle, 2012; Pastawski et al., 2015), which leads to the exciting development of machine learning holography (You et al., 2018; Hashimoto et al., 2018; Hashimoto, 2019; Akutagawa et al., 2020; Hashimoto et al., 2020).

## 3 METHODS

**Flow-based generative models.** Flow-based generative models are a family of generative models with tractable distributions, which allows efficient sampling and exact evaluation of the probability density (Dinh et al., 2015; 2017; Kingma & Dhariwal, 2018; Chen et al., 2019). The key idea is to build a bijective map $G(\boldsymbol{z}) = \boldsymbol{x}$ between visible variables $\boldsymbol{x}$ and latent variables $\boldsymbol{z}$. Visible variables $\boldsymbol{x}$ are the data that we want to generate, which may follow a complicated probability distribution. And latent variables $\boldsymbol{z}$ usually have simple distribution that can be easily sampled, for example the i.i.d. Gaussian distribution. In this way, the data can be efficiently generated by first sampling $\boldsymbol{z}$ and mapping them to $\boldsymbol{x}$ through $\boldsymbol{x} = G(\boldsymbol{z})$. In addition, we can get the probability associated with each data sample $\boldsymbol{x}$,

$$\log p_X(\boldsymbol{x}) = \log p_Z(\boldsymbol{z}) - \log \left| \frac{\partial G(\boldsymbol{z})}{\partial \boldsymbol{z}} \right|. \tag{1}$$

The bijective map $G(\boldsymbol{z}) = \boldsymbol{x}$ is usually composed as a series of bijectors, $G(\boldsymbol{z}) = G_1 \circ G_2 \circ \cdots \circ G_n(\boldsymbol{z})$, such that each bijector layer $G_i$ has a tractable Jacobian determinant and can be inverted efficiently. The two key ingredients in flow-based models are the design of the bijective map $G$ and the choice of the prior distribution $p_Z(\boldsymbol{z})$.

**Structure of RG-Flow networks.** Much of the prior research has focused on designing more powerful bijective blocks for the generator $G$ to improve its expressive power and to achieve better approximations of complicated probability distributions. Here, we focus on designing the architecture that arranges the bijective blocks in a hierarchical structure to separate features of different [1] scales in the data and to disentangle latent representations.

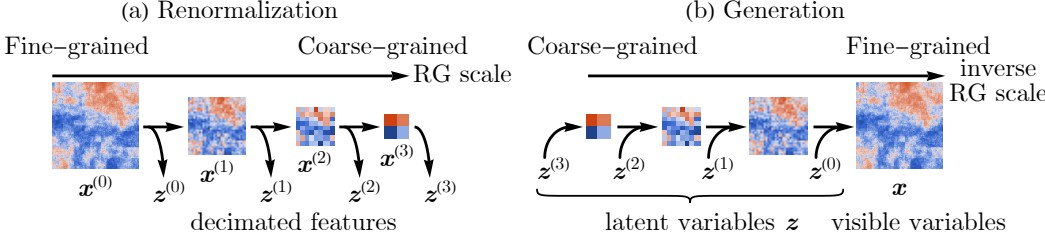

Figure 1: (a) The forward RG transformation splits out decimated features at different scales. (b) The inverse RG transformation generates the fine-grained image from latent variables.

Our design is motivated by the idea of RG in physics, which progressively separates the coarse-grained data statistics from fine-grained statistics by local transformations at different scales. Let $\boldsymbol{x}$ be the visible variables, or the input image (level-0), denoted as $\boldsymbol{x}^{(0)} \equiv \boldsymbol{x}$. A step of the RG transformation extracts the coarse-grained information $\boldsymbol{x}^{(1)}$ to send to the next layer (level-1), and splits out the rest of fine-grained information as auxiliary variables $\boldsymbol{z}^{(0)}$. The procedure can be

described by the following recursive equation (at level-$h$ for example),

$$\boldsymbol{x}^{(h+1)}, \boldsymbol{z}^{(h)} = R_h(\boldsymbol{x}^{(h)}), \tag{2}$$

which is illustrated in Fig. 1(a), where $\dim(\boldsymbol{x}^{(h+1)}) + \dim(\boldsymbol{z}^{(h)}) = \dim(\boldsymbol{x}^{(h)})$, and the RG transformation $R_h$ can be made invertible. At each level, the transformation $R_h$ is a local bijective map, which is constructed by stacking trainable bijective blocks. We will specify its details later. The split-out information $\boldsymbol{z}^{(h)}$ can be viewed as latent variables arranged at different scales. Then the inverse RG transformation $G_h \equiv R_h^{-1}$ simply generates the fine-grained image,

$$\boldsymbol{x}^{(h)} = R_h^{-1}(\boldsymbol{x}^{(h+1)}, \boldsymbol{z}^{(h)}) = G_h(\boldsymbol{x}^{(h+1)}, \boldsymbol{z}^{(h)}). \tag{3}$$

The highest-level image $\boldsymbol{x}^{(h_L)} = G_{h_L}(\boldsymbol{z}^{(h_L)})$ can be considered as generated directly from latent variables $\boldsymbol{z}^{(h_L)}$ without referring to any higher-level coarse-grained image, where $h_L = \log_2 L - \log_2 m$, for the original image of size $L \times L$ with local transformations acting on kernel size $m \times m$. Therefore, given the latent variables $\boldsymbol{z} = \{\boldsymbol{z}^{(h)}\}$ at all levels $h$, the original image can be restored by the following nested maps, as illustrated in Fig. 1(b),

$$\boldsymbol{x} \equiv \boldsymbol{x}^{(0)} = G_0(G_1(G_2(\cdots, \boldsymbol{z}^{(2)}), \boldsymbol{z}^{(1)}), \boldsymbol{z}^{(0)}) \equiv G(\boldsymbol{z}), \tag{4}$$

where $\boldsymbol{z} = \{\boldsymbol{z}^0, \cdots, \boldsymbol{z}^{h_L}\}$. RG-Flow is a flow-based generative model that uses the above composite bijective map $G$ as the generator.

To model the RG transformation, we arrange the bijective blocks in a hierarchical network architecture. Fig. 2(a) shows the side view of the network, where each green or yellow block is a local bijective map. Following the notation of MERA networks, the green blocks are the *disentanglers*, which reparametrize local variables to reduce their correlations, and the yellow blocks are the *decimators*, which separate the decimated features out as latent variables. The blue dots on the bottom are the visible variables $\boldsymbol{x}$ from the data, and the red crosses are the latent variables $\boldsymbol{z}$. We omit color channels of the image in the illustration, since we keep the number of color channels unchanged through the transformation.

Fig. 2(b) shows the top-down view of a step of the RG transformation. The green/yellow blocks (disentanglers/decimators) are interwoven on top of each other. The covering area of a disentangler or decimator is defined as the kernel size $m \times m$ of the bijector. For example, in Fig. 2(b), the kernel size is $4 \times 4$. After the decimator, three fourth of the degrees of freedom are decimated into latent variables (red crosses in Fig. 2(a)), so the edge length of the image is halved.

As a mathematical description, for the single-step RG transformation $R_h$, in each block $(p, q)$ labeled by $p, q = 0, 1, \ldots, \frac{L}{2^h m} - 1$, the mapping from $\boldsymbol{x}^{(h)}$ to $(\boldsymbol{x}^{(h+1)}, \boldsymbol{z}^{(h)})$ is given by

$$\left\{\boldsymbol{y}^{(h)}_{2^h(mp+\frac{m}{2}+a, mq+\frac{m}{2}+b)}\right\}_{(a,b)\in\square_m^1} = R_h^{\text{dis}}\left(\left\{\boldsymbol{x}^{(h)}_{2^h(mp+\frac{m}{2}+a, mq+\frac{m}{2}+b)}\right\}_{(a,b)\in\square_m^1}\right)$$

$$\left\{\boldsymbol{x}^{(h+1)}_{2^h(mp+a, mq+b)}\right\}_{(a,b)\in\square_m^2}, \left\{\boldsymbol{z}^{(h)}_{2^h(mp+a, mq+b)}\right\}_{(a,b)\in\square_m^1/\square_m^2} = R_h^{\text{dec}}\left(\left\{\boldsymbol{y}^{(h)}_{2^h(mp+a, mq+b)}\right\}_{(a,b)\in\square_m^1}\right), \tag{5}$$

where $\square_m^k = \{(ka, kb) \mid a, b = 0, 1, \ldots, \frac{m}{k} - 1\}$ denotes the set of pixels in a $m \times m$ square with stride $k$, and $\boldsymbol{y}$ is the intermediate result after the disentangler but not the decimator. The notation $\boldsymbol{x}^{(h)}_{(i,j)}$ stands for the variable (a vector of all channels) at the pixel $(i, j)$ and at the RG level $h$ (similarly for $\boldsymbol{y}$ and $\boldsymbol{z}$). The disentanglers $R_h^{\text{dis}}$ and decimators $R_h^{\text{dec}}$ can be any bijective neural network. Practically, We use the coupling layer proposed in the Real NVP networks (Dinh et al., 2017) to build them, with a detailed description in Appendix A. By specifying the RG transformation $R_h = R_h^{\text{dec}} \circ R_h^{\text{dis}}$ above, the generator $G_h \equiv R_h^{-1}$ is automatically specified as the inverse transformation.

**Training objective.** After decomposing the statistics into multiple scales, we need to make the latent features decoupled. So we assume that the latent variables $\boldsymbol{z}$ are independent random variables, described by a factorized prior distribution

$$p_Z(\boldsymbol{z}) = \prod_l p(z_l), \tag{6}$$

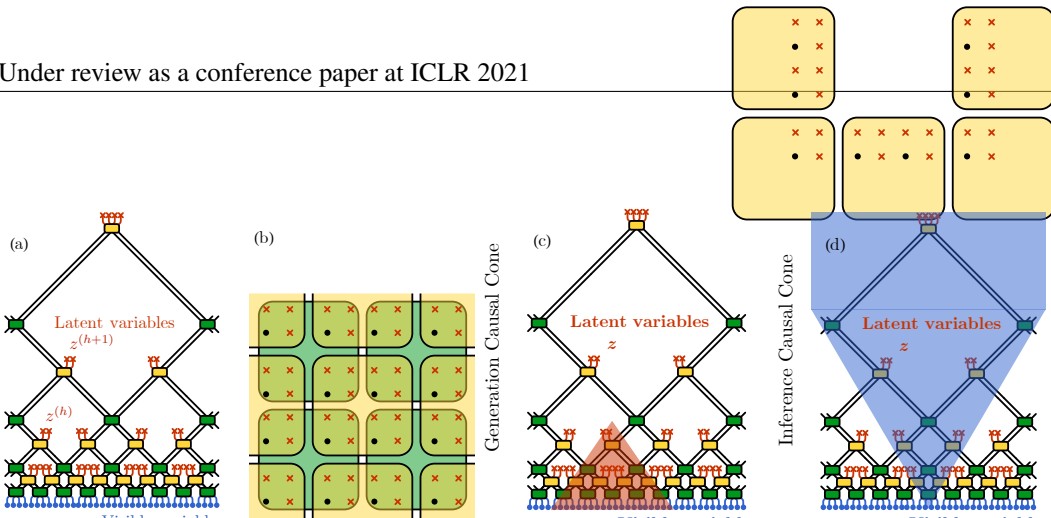

Figure 2: Subplot (a) shows the side view of the network. Green/Yellow blocks denote the disentanglers/decimators, which are bijective maps. Subplot (b) shows the top-down view of the network. The red area in the subplot (c) illustrates the generation causal cone for a latent variable. The blue area in the subplot (d) illustrates the inference causal cone for a visible variable.

where $l$ labels every element in $\boldsymbol{z}$, including the RG level, the pixel position and the channel. This prior gives the network the incentive to minimize the mutual information between latent variables. This *minimal bulk mutual information* (minBMI) principle was previously proposed to be the information theoretic principle that defines the RG transformation (Li & Wang (2018); Hu et al. (2020)).

Starting from a set of independent latent variables $\boldsymbol{z}$, the generator $G$ should build up correlations locally at different scales, such that the multi-scale correlation structure can emerge in the resulting image $\boldsymbol{x}$ to model the correlated probability distribution of the data. To achieve this goal, we should maximize the log likelihood for $\boldsymbol{x}$ drawn from the data set. The loss function to minimize reads

$$\mathcal{L} = -\mathbb{E}_{\boldsymbol{x} \sim p_{\text{data}}(\boldsymbol{x})} \log p_X(\boldsymbol{x}) = -\mathbb{E}_{\boldsymbol{x} \sim p_{\text{data}}(\boldsymbol{x})} \left( \log p_Z(R(\boldsymbol{x})) + \log \left| \frac{\partial R(\boldsymbol{x})}{\partial \boldsymbol{x}} \right| \right), \qquad (7)$$

where $R(\boldsymbol{x}) \equiv G^{-1}(\boldsymbol{x}) = \boldsymbol{z}$ denotes the RG transformation, which contains trainable parameters. By optimizing the parameters, the network learns the optimal RG transformation from the data.

**Receptive fields of latent variables.** Due to the nature of local transformations in our hierarchical network, we can define the generation causal cone for a latent variable to be the affected area when that latent variable is changed. This is illustrated as the red cone in Fig. 2(c).

To visualize the latent space representation, we define the *receptive field* for a latent variable $z_l$ as

$$\text{RF}_l = \mathbb{E}_{\boldsymbol{z} \sim p_Z(\boldsymbol{z})} \left| \frac{\partial G(\boldsymbol{z})}{\partial z_l} \right|_c, \qquad (8)$$

where $| \cdot |_c$ denotes the 1-norm on the color channel. The receptive field reflects the response of the generated image to an infinitesimal change of the latent variable $z_l$, averaged over $p_Z(\boldsymbol{z})$. Therefore, the receptive field of a latent variable is always contained in its generation causal cone. Higher-level latent variables have larger receptive fields than those of the lower-level ones. Especially, if the receptive fields of two latent variables do not overlap, which is often the case for lower-level latent variables, they automatically become disentangled in the representation.

**Image inpainting and error correction.** Another advantage of the network locality can be demonstrated in the inpainting task. Similar to the generation causal cone, we can define the *inference causal cone* shown as the blue cone in Fig. 2(d). If we perturb a pixel at the bottom of the blue cone, all the latent variables within the blue cone will be affected, whereas the latent variables outside the cone cannot be affected. An important property of the hyperbolic tree-like network is that the higher level contains exponentially fewer latent variables. Even though the inference causal cone is expanding as we go into higher levels, the number of latent variables dilutes exponentially as well, resulting in a constant number of latent variables covered by the inference causal cone on each level. Therefore, if a small local region on an image is corrupted, only $O(\log L)$ latent variables need to be modified, where $L$ is the edge length of the entire image. While for globally connected networks, all $O(L^2)$ latent variables have to be varied.

**Sparse prior distribution.** We have chosen to hard-code the RG information principle by using a factorized prior distribution, i.e. $p_Z(\boldsymbol{z}) = \prod_l p(z_l)$. The common practice is to choose $p(z_l)$ to be the standard Gaussian distribution, which is spherical symmetric. If we apply any rotation to $\boldsymbol{z}$, the distribution will remain the same. Therefore, we cannot avoid different features from being mixed under the arbitrary rotation.

To overcome this issue, we use an anisotropic sparse prior distribution for $p_Z(\boldsymbol{z})$. In our implementation, we choose the Laplacian distribution $p(z_l) = \frac{1}{2b} \exp(-|z_l|/b)$, which is sparser compared to Gaussian distribution and breaks the spherical symmetry of the latent space. In Appendix E, we show a two-dimensional pinwheel example to illustrate this intuition. This heuristic method will encourage the model to find more semantically meaningful representations by breaking the spherical symmetry.

## 4 EXPERIMENTS

**Synthetic multi-scale datasets.** To illustrate RG-Flow's ability to disentangle representations at different scales and spatially separated representations, we propose two synthetic datasets with multi-scale features, named MSDS1 and MSDS2. Their samples are shown in Appendix B. In each image, there are 16 ovals with different colors and orientations. In MSDS1, all ovals in an image have almost the same color, while their orientations are randomly distributed. So the color is a global feature in MSDS1, and the orientation is a local feature. In MSDS2, on the contrary, the orientation is a global feature, and the color is a local one.

We implement RG-Flow as shown in Fig. 2. After training, we find that RG-Flow can easily capture the characteristics of those datasets. Namely, the ovals in each image from MSDS1 have almost the same color; and from MSDS2, the same orientation. Especially, in Fig. 3, we plot the effect of varying latent variables at different levels, together with their receptive fields. For MSDS1, if we vary a high-level latent variable, the color of the whole image will change, which shows that the network has captured the global feature of the dataset. And if we vary a low-level latent variable, the orientation of only the corresponding one oval will change. As the ovals are spatially separated, the low-level representation of different ovals is disentangled. Similarly, for MSDS2, if we vary a high-level latent variable, the orientations of all ovals will change. And if we vary a low-level latent variable, the color of only the corresponding one oval will change.

For comparison, we also trained Real NVP on our synthetic datasets. We find that Real NVP fails to learn the global and local characteristics of those datasets. Details can be found in Appendix B.

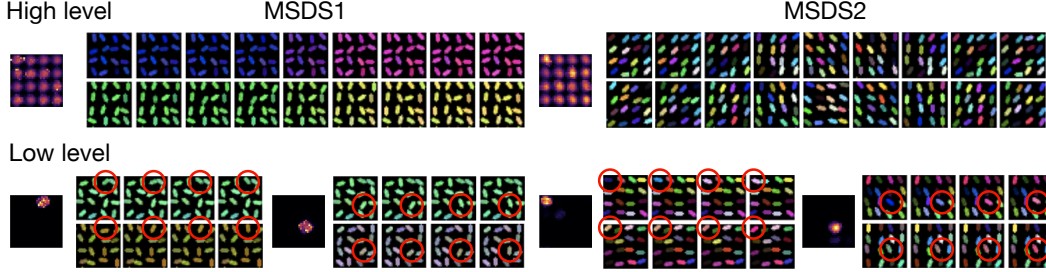

Figure 3: Multi-scale latent representations for MSDS1 and MSDS2.

**Human face dataset.** Next, we apply RG-Flow to more complicated multi-scale datasets. Most of our experiments use the human face dataset CelebA (Liu et al., 2015), and we crop and scale the images to $32 \times 32$ pixels. Details of the network and the training procedure can be found in Appendix A. Experiments on other datasets, such as CIFAR-10 (Krizhevsky et al.), and quantitative evaluations can also be found in Appendix G.

After training, the network learns to progressively generate finer-grained images, as shown in Fig. 4(a). The colors in the coarse-grained images are not necessarily the same as those at the same positions in the fine-grained images, because there is no constraint to prevent the RG transformation from mixing color channels.

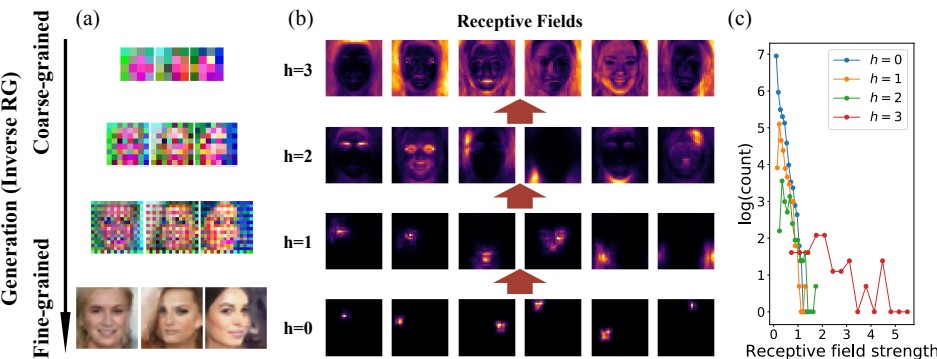

Figure 4: Subplot (a) shows the progressive generation of images during the inverse RG. Subplot (b) shows some receptive fields of latent variables from low level to high level. The strength of each receptive field is rescaled to one for better visualization. Subplot (c) shows the statistics of the receptive fields' strength.

**Receptive fields.** To visualize the latent space representation, we calculate the receptive field for each latent variable, and list some of them in Fig. 4(b). We can see the receptive size is small for low-level variables and large for high-level ones, as indicated from the generation causal cone. In the lowest level ($h = 0$), the receptive fields are merely small dots. In the second lowest level ($h = 1$), small structures emerge, such as an eyebrow, an eye, a part of hair, etc. In the middle level ($h = 2$), we can see eyebrows, eyes, forehead bang structure emerge. In the highest level ($h = 3$), each receptive field grows to the whole image. We will investigate those explainable latent representations in the next section. For comparison, we show receptive fields of Real NVP in Appendix C. Even though Real NVP has multi-scale structure, since it is not locally constrained, semantic representations at different scales do not emerge.

**Learned features on different scales.** In this section, we show that some of these emergent structures correspond to explainable latent features. Flow-based generative model is the *maximal encoding procedure*, because the core of flow-based generative models is the bijective maps, and they preserves the dimensionality before and after the encoding. Usually, the images in the dataset live on a low dimensional manifold, and we do not need to use all the dimensions to encode such data. In Fig. 4(c) we show the statistics of the strength of receptive fields. We can see most of the latent variables have receptive fields with relatively small strength, meaning that if we change the value of those latent variables, the generated images will not be affected much. We focus on those latent variables with receptive field strength greater than one, which have visible effects on the generated images. We use $h$ to label the RG level of latent variables, for example, the lowest-level latent variables have $h = 0$, whereas the highest-level latent variables have $h = 4$. In addition, we will focus on $h = 1$ (low level), $h = 2$ (mid level), $h = 3$ (high level) latent variables. There are a few latent variables with $h = 0$ that have visible effects, but their receptive fields are only small dots with no emergent structures.

For high-level latent representations, we found in total 30 latent variables that have visible effects, and six of them are identified with disentangled and explainable meanings. Those factors are gender, emotion, light angle, azimuth, hair color, and skin color. In Fig. 5(a), we plot the effect of varying those six high-level variables, together with their receptive fields. For the mid-level latent representations, we plot the four leading variables together with their receptive fields in Fig. 5(b), and they control eye, eyebrow, upper right bang, and collar respectively. For the low-level representations, some leading variables control an eyebrow and an eye as shown in Fig. 5(c). We see them achieve better disentangled representations when their receptive fields do not overlap.

**Image mixing in scaling direction.** Given two images $\boldsymbol{x}_A$ and $\boldsymbol{x}_B$, the conventional image mixing takes a linear combination between $\boldsymbol{z}_A = G^{-1}(\boldsymbol{x}_A)$ and $\boldsymbol{z}_B = G^{-1}(\boldsymbol{x}_B)$ by $\boldsymbol{z} = \lambda \boldsymbol{z}_A + (1 - \lambda)\boldsymbol{z}_B$ with $\lambda \in [0, 1]$ and generates the mixed image from $\boldsymbol{x} = G(\boldsymbol{z})$. In our model, latent variables $\boldsymbol{z}$ is coordinated by the pixel position $(i, j)$ and the RG level $h$. The direct access of the latent variable

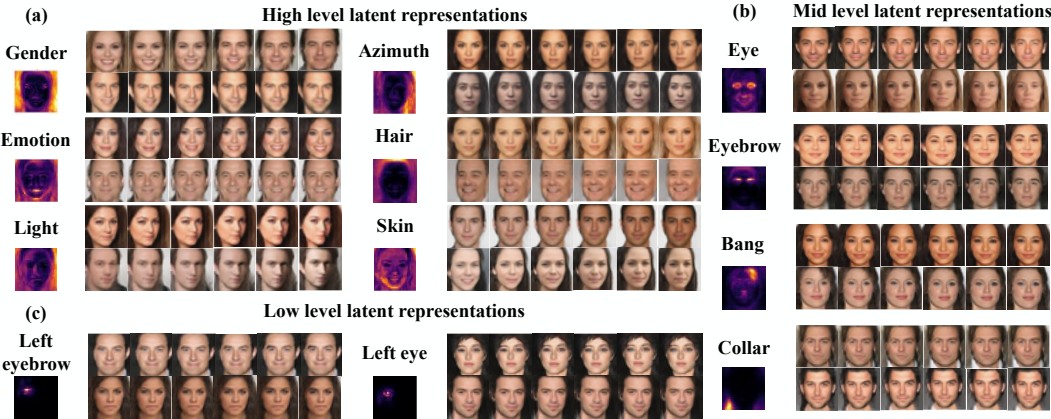

Figure 5: Semantic factors found on different levels.

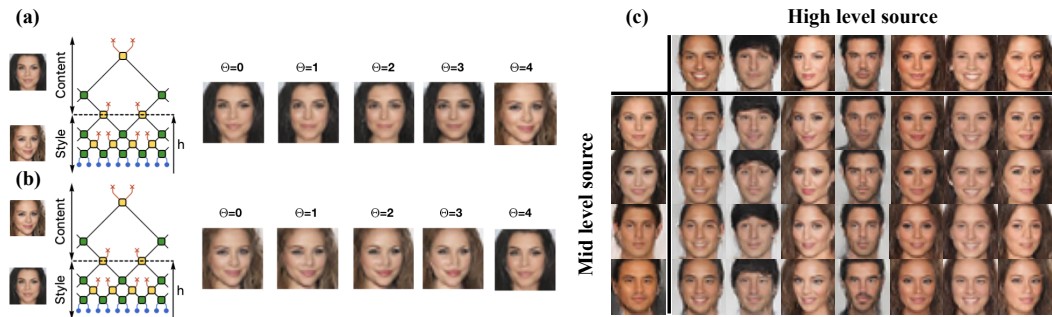

Figure 6: Image mixing in the hyperbolic tree-like latent space.

$z_{(i,j)}^{(h)}$ at each point enables us to mix the latent variables in a different manner, which may be dubbed as a "hyperbolic mixing". We consider mixing the large-scale (high-level) features of $x_A$ and the small-scale (low-level) features of $x_B$ by combining their corresponding latent variables via

$$z^{(h)} = \begin{cases} z_A^{(h)}, & \text{for } h \geq \Theta, \\ z_B^{(h)}, & \text{for } h < \Theta, \end{cases} \qquad (9)$$

where $\Theta$ serves as a dividing line of the scales. As shown in Fig. 6(a), as we change $\Theta$ from 0 to 3, more low-level information in the blonde-hair image is mixed with the high-level information of the black-hair image. Especially when $h = 3$, we see the mixed face have similar eyes, nose, eyebrows, and mouth as the blonde-hair image, while the high-level information, such as face orientation and hair color, is taken from the black-hair image. In addition, this mixing is not symmetric under the interchange of $z_A$ and $z_B$, see Fig. 6(b) for comparison. This hyperbolic mixing achieves the similar effect of StyleGAN (Karras et al., 2019; 2020) that we can take mid-level information from an image and mix it with the high-level information of another image. In Fig. 6(c), we show more examples of mixing faces.

**Image inpainting and error correction.** The existence of the inference causal cone ensures that at most $O(\log L)$ latent variables will be affected, if we have a small local corrupted region to be inpainted. In Fig. 7, we show that RG-Flow can faithfully recover the corrupted region (marked as red) only using latent variables locating inside the inference causal cone, which are around one third of all latent variables. For comparison, if we randomly pick the same number of latent variables to modify in Real NVP, it fails to inpaint as shown in Fig. 7 (Constrained Real NVP). To achieve the recovery of similar quality in Real NVP, as shown in Fig. 7 (Real NVP), all latent variables need to be modified, which are of $O(L^2)$ order. See Appendix F for more details about the inpainting task and its quantitative evaluations.

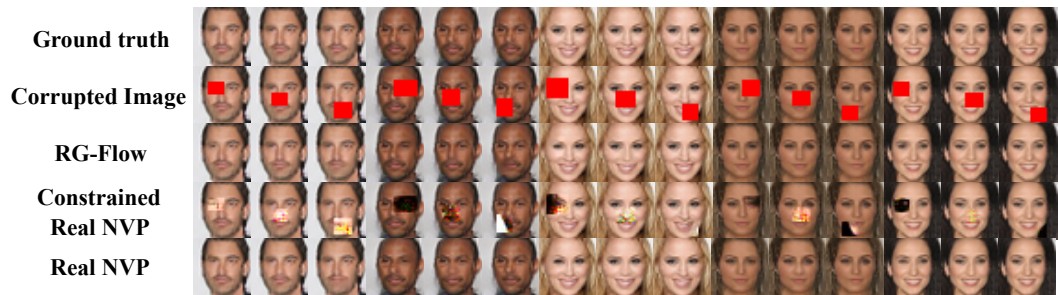

Figure 7: Inpainting locally corrupted images.

## 5 DISCUSSION AND CONCLUSION

In this paper, we combined the ideas of renormalization group and sparse prior distribution to design RG-Flow, a probabilistic flow-based generative model. This versatile architecture can be incorporated with any bijective map to achieve an expressive flow-based generative model. We have shown that RG-Flow can separate information at different scales and encode them in latent variables living on a hyperbolic tree. To visualize the latent representations in RG-Flow, we defined the receptive fields for flow-based models in analogy to that in CNN. Taking CelebA dataset as our main example, we have shown that RG-Flow will not only find high-level representations, but also mid-level and low-level ones. The receptive fields serve as a visual guidance for us to find explainable representations. In contrast, the semantic representations of mid-level and low-level structures do not emerge in globally connected multi-scale flow models, such as Real NVP. We have also shown that the latent representations can be mixed at different scales, which achieves an effect similar to style mixing.

In our model, if the receptive fields of two latent representations do not overlap, they are naturally disentangled. For high-level representations, we propose to utilize a sparse prior to encourage disentanglement. We find that if the dataset only contains a few high-level factors, such as the 3D Chair dataset (Aubry et al., 2014) shown in Appendix G, it is hard to find explainable high-level disentangled representations, because of the redundant nature of the encoding in flow-based models. Incorporating information theoretic criteria to disentangle high-level representations in the redundant encoding procedure will be an interesting future direction.

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
