# OpenReview forum: "RG-Flow: A hierarchical and explainable flow model based on renormalization group and sparse prior"
_ICLR.cc/2021/Conference — Reject_

### Official Review · AnonReviewer2 · 2020-10-26

**Rating:** 5
**Confidence:** 3

**Review:**

Summary:
The submission suggests a generative model with a normalizing flow based on a renormalization group idea. This gives rise to a multi-scale latent representation by arranging bijective transformations hierarchically with depth scaling logarithmically with the input dimension. Extensive experiments on CelebA are conducted, illustrating the latent feature on different scales and image inpainting experiments.

Positives:
The paper includes substantive experiments that (i) visualize the latent features on the different hierarchical levels, (ii) illustrate mixing of latent variables across different levels, (iii) show good performance in error correction.
The proposed RG-Flow network architecture is very interesting indeed and is new for the considered tasks as far as I am aware. The complexity of the approach can improve from O(L^2) for previous normalizing flows for image inpainting to O(log L). This is a very significant improvement (but it is not clear to me how well the architecture performs for corruptions that are not just in a single local region).

Negatives:
Similar flow architectures have been used before in different contexts (mostly physics) as also mentioned in the paper, but I feel that it is not so clear how the proposed normalizing flow model differs from previous work with a renormalisation group (NeuralRG). The role of symmetric priors for disentangled representations has been studied previously (using information theoretical ideas, for VAEs etc) and I also feel that it should be clarified how the sparse Laplace prior differs from previous work.

Recommendation:
This appears a borderline submission for me, but my confidence for assessing it is not that high. The experimental section is solid and I would consider increasing the score if the differences to previous work are made clearer.

Comments:
Does the choice of the temperature for the prior has a large influence on the learned features?
How does the computational complexity and running time of the RG architecture compare with other flow based generative models like Glow/(plain) RNVP?

---

> ### Author Response · Authors · 2020-11-24
> **Response to Reviewer 2**
>
> Thanks for reviewing our paper, and pointing out some questions!
>
> ---
>
> 1. "Similar flow architectures have been used before in different contexts (mostly physics) as also mentioned in the paper, but I feel that it is not so clear how the proposed normalizing flow model differs from previous work with a renormalisation group (NeuralRG)."
>
> The original NeuralRG is designed especially for physical systems for Monte Carlo simulations (Li & Wang, 2018) or study of latent space geometry (Hu et al., 2020). However, it cannot be directly applied to complicated natural images and there has not been results on natural image datasets. Through our trials, directly applying those structures to natural images leads to abysmal results. To apply the idea of RG to more complicated datasets, we implement more expressive local bijective maps involving deep residue blocks, swish activation, and weight normalization etc. Our implementation makes RG-Flow capable of generating natural images. We are the first to thoroughly discuss the multi-scale latent representations under the RG transformation. An additional contribution is that we define the receptive fields of flow-based models and generation/inference causal cones for better theoretical understanding of RG-Flow.
>
> In the updated "Experiments" section and Appendix B, we add the discussion of our synthetic multiscale datasets MSDS1 and MSDS2, which illustrate that RG-Flow can separate high-level and low-level representations, while other flow-based models like Real NVP cannot achieve that. For the human face dataset, we also add the comparison with Real NVP in Appendix C.
>
> ---
>
>
> 2. "The role of symmetric priors for disentangled representations has been studied previously (using information theoretical ideas, for VAEs etc) and I also feel that it should be clarified how the sparse Laplace prior differs from previous work."
>
>
> In beta-TCVAE and FactorVAE, the high level disentanglement is encouraged by independence of latent variables. In RG-Flow, the information bottleneck is achieved by RG and all latent variables are independent. To make a step further, we use the sparse prior distribution as a heuristic method to encourage the disentanglement of latent variables, which may have overlapping receptive fields when not disentangled. The intuition is that the sparse distribution breaks the rotational symmetry of latent space, which is visualized in Appendix E.
>
> However, we are aware that the disentanglement of high-level representations in an unsupervised manner is an ill-defined problem, which is also the main conclusion of [1]. So it is not our focus to disentangle the high-level representations. In contrast, we find the disentanglement of mid-level and low-level representations in RG-Flow is well-defined. If the receptive fields of two latent variables are not overlapping, we can say they are disentangled. This is discussed for our newly added synthetic toy datasets, and also for the human face dataset.
>
> [1] Francesco Locatello, et al., Challenging Common Assumptions in the Unsupervised Learning of Disentangled Representations. (2019)
>
>
> ---
>
>
> 3. "Does the choice of the temperature for the prior has a large influence on the learned features?"
>
> During training we always set the variance of the prior distribution to 1, that is $T = 1$, so it will not influence the learned features. We only lower the temperatures for better visual quality when generating images from the trained network, as discussed in Appendix D.
>
> ---
>
> 4. "How does the computational complexity and running time of the RG architecture compare with other flow based generative models like Glow/(plain) RNVP?"
>
> In the inpainting task, we showed that theoretically RG-Flow only need to do optimization on $O(\log L)$ latent variables, while other flow-based generative models with global connections need to do so on all $L^2$ latent variables.
>
> For now, the $O(\log L)$ advantage is more of a theoretical one than a practical one. The advantage in asymptotic complexity will be significant when L is large, but because of our limited computation resources, we can only implement L up to 32.
>
> ---
>
> 5. "But it is not clear to me how well the architecture performs for corruptions that are not just in a single local region."
>
> The $O(\log L)$ advantage is only valid if the corrupted region is small and local compared to the whole image. Otherwise, if the artifacts are not local, such as the salt-and-pepper type of noise distributed over the whole image, the union of the inference casual cones of those artifacts will cover almost all latent variables, and we need to optimize all of them. Even in that case, the quality of the inpainted image is comparable to other generative models like Real NVP.

---

### Official Review · AnonReviewer3 · 2020-10-27
**Reasonable extension of flow-based method but contributions not significant**

**Rating:** 5
**Confidence:** 4

**Review:**

## Summary

The paper proposes a method, named as RG-flow, which combines the ideas of Renormalization group (RG) and flow-based models. The RG is applied to separate signal statistics of different scales in the input distribution and flow-based idea represents each scale information in its latent variables with sparse prior distribution. Inspired by receptive field from CNNs, the authors visualize the latent space representation, which reveals the progressive semantics at different levels as the instinctive expectation.

## Strengths

+. The proposed method presents a simple method to combine renormalization group and flow-based models.

+. The visualized receptive field of latent space representation shows meaningful semantics at different levels, which is verified on CelebA.

## Weaknesses/Concerns

-. It seems like a combination of existing methods, including Flow-based methods, renormalization group and receptive field. Compared with the two mentioned NeuralRG papers(Li & Wang, 2018; Hu et al., 2020), what are the differences for the renormalization group part? The main contributions of the proposed method are not so clear.

-. In the flow part, the high-dimensional visual variables are modeled by a sparse prior distribution, while the visual results (e.g., Figure 4, 5, 6 in the main paper) are with limited diversity. Does mode collapse happen during modeling the distribution?

-. It could be too strong that enforces the high-dimensional features at different layers to follow a same sparse prior distribution. Though the proposed method achieve acceptable results on CelebA which is an almost aligned face dataset, it fails to model the images in the wild (e.g., CIFAR-10 on Figure 8 in supplementary). Does it mean the proposed method works only on aligned/structured scenes? Specially, compared with the style-based generator[1,2], where the sparse prior distribution is enforced only in the input z latent space, I don’t find superiorities of the proposed method.

[1] A Style-Based Generator Architecture for Generative Adversarial Networks, CVPR 2019.

[2] StyleFlow: Attribute-conditioned Exploration of StyleGAN-Generated Images using Conditional Continuous Normalizing Flows, arXiv 2020

-. Given the visualization methods can have a great effect on the results, Fig. 9 is a bit weak to show the advantages of breaking the spherical symmetry of the style latent. What are the clusters in latent space, visual space and target?

-. The experiments are conducted on a low-resolution setting (i.e., 32x32 on CelebA). Does it work on higher resolution?

-. There is no qualitative and quantitative evalution to compare the performance or show the improvement gained by the proposed method.

-. In order to verify renormalization group helps on learning more meaningful semantics at different levels, it's better to show an ablation study to compare the visualization of receptive field between w/ RG and w/o RG.

-. What’s the meaning of y in equation (5)?


## Overall Recommendation

Based on the above weaknesses/concerns, I rate the paper "Marginally below acceptance threshold".

---

> ### Author Response · Authors · 2020-11-24
> **Response to Reviewer 3**
>
> Thanks for reviewing our paper and pointing out some questions!
>
> ---
>
> 1. ".. Compared with the two mentioned NeuralRG papers .. the differences for the renormalization group part?"
>
> For this question, please kindly refer to our general comment (1) and (2).
>
> ---
>
>
> 2. ".. the visual results are with limited diversity. Does mode collapse happen ..?"
>
> We have added more samples in Appendix G, Fig. 13 to show the diversity of generated images, including different genders, skin tones, face orientations, expressions and more factors.
>
> ---
>
> 3. ".. too strong that enforces .. features at different layers .. a same sparse prior"
>
> It is also a choice to optimize the variance of the prior distribution of each latent variable, or add more optimizable parameters into the priors. Currently, we are not doing so, because the bijective maps can already scale the variance or change the shape of the priors.
>
> ---
>
> 4. "..does proposed method work only on aligned/structured scenes?"
>
> In our network, we do not impose anything that favors aligned datasets. So it should also work on non-aligned datasets. We think that images in the wild naturally follow a multi-scale structure, which is captured by RG. Our implementation is largely based on Real NVP, which set the expressiveness of the bijective mapping. For natural dataset like CIFAR-10, we find RG-Flow achieves even better performance than Real NVP, please see Appendix G in the revision for the quantitative comparison. We feel this is a strong indication that RG-Flow captures the multiscale characteristics in CIFAR-10. We expect that if more advanced bijective maps are used, such as invertible convolutions and i-ResNet, the performance of RG-Flow can be further improved.
>
> ---
>
> 5. "compared with the style-based generator[1,2] .. I don’t find superiorities of the proposed method."
>
> Thanks for pointing these interesting papers of style-based generative models, which can also perform the mixture of human face. The advantage of RG-Flow is that its mid-level and low-level representations have clear semantic meanings, and they are guaranteed to be disentangled since they do not overlap spatially. E.g. in the human face dataset, the mid-level factor of eyes will only change the eyes and not affect the hair, since they do not overlap.
>
> ---
>
> 6. ".. Fig. 12 is a bit weak to show the advantages of breaking the spherical symmetry .. What are the clusters in latent space, visual space .."
>
> We agree that breaking symmetry of latent space is only a heuristic method. Practically we find it works well. Fig. 12 is an intuitive illustration. The different colors are not for clustering. Instead, they are used to visualize the correspondence between points in latent and visible spaces. This is an convenient visualization of the diffeomorphism. In the latent space, points in different quadrants are given different colors.
>
> ---
>
> 7. "experiments .. on a low-resolution setting (i.e., 32x32 on CelebA) .. higher resolution?"
>
> We expect it will work on higher resolution images with more expressive local bijective maps as the performance of RG-Flow depends largely on the family of local bijective maps.
>
> ---
>
> 8. "no .. evaluation to compare the performance .."
>
> For the qualitative comparison, our discussion of Fig. 4 and Appendix C shows that RG-Flow can produce semantically meaningful representations from low level to high level which is theoretically guaranteed. In contrast, Real NVP cannot produce semantically meaningful mid-level or low-level representations.
>
> In the revision, we create two synthetic toy datasets with multi-scale features, named MSDS1 and MSDS2. This is shown in "Experiments" section and in Appendix B, that RG-Flow can separately capture high-level and low-level features, while Real NVP fails to do so.
>
> For the quantitative evaluation, there are already BPD scores to assess the quality and diversity of generated images in Appendix G. We have added FID as another metric. FID score is not widely used to assess flow-based models, because flow-based models can directly give the probability density of each generated sample, whereas GAN cannot. BPD may be a more fair metric on datasets of specific fields like human faces and synthetic datasets, and it avoids the bias from the knowledge encoded in Inception-v3.
>
> In Appendix F, we also add the metrics of PSNR to quantitatively assess the quality of inpainted images with limited size of latent space for optimization.
>
> ---
>
> 9. "to verify renormalization group helps .. semantics at different levels .. better to .. compare .. receptive field between w/ RG and w/o RG."
>
> Sure! In Appendices B and C of the updated text, we add the comparison of RG-Flow and Real NVP as an ablation study. Except for RG, the structure of RG-Flow is largely similar to Real NVP.
>
> ---
>
> 10. "What’s the meaning of y in equation (5)?"
>
> y is the intermediate result after transforming by the disentangler (green block in Fig. 2(b)), but not the decimator (yellow block).

---

### Official Review · AnonReviewer4 · 2020-10-27
**RG-Flow Review**

**Rating:** 6
**Confidence:** 3

**Review:**

The paper proposes a new architecture for flow based generative models. The model imposes a hierarchical structure over information at different scales. The paper shows that the hierarchical structure results in disentangled features at different levels of abstractions. The paper claims that the approach is based on the renormalization group


Strengths:
+ The overall structure of the approach is intuitive, and the writing/figures are understandable
+ I like how the approach naturally produces disentangled features at different levels of abstraction.
+ Figure 9 demonstrates how the sparse prior leads to better disentanglement than the Gaussian prior. Due to the rotation invariance of the Gaussian prior, the hidden dimensions of other flow based generative models like Real NVP don't have grounded semantic meaning.
+ The hierarchical approach shows a small improvement over


Weaknesses:
- (Intro/related work) There should be more discussion how this paper relates to Neural Network Renormalization Group (Li & Wang). Some more background/examples of the renormalization group in physics / meaning of the disentangler/decimator would be helpful

- There is no evaluation of the image inpainting performance. Some other papers (Structured Output Learning with Conditional Generative Flows ,Lu & Huang) use PSNR.
- Constrained NVP seems to be too weak a baseline for the image inpainting task. Are there more reasonable comparisons? Since the paper claims a log(L) vs L^2 improvement, timing or flop counts should be used to compare to Real NVP.
- The paper provides limited quantitative results and comparisons. Only qualitative comparisons are provided on the in-painting task. There are no quantitative results provided in the main paper, and only results on CelebA, CIFAR-10, and 3D chair in the appendix.  There is no discussion of the 3D chair experiment in the appendix.

Some minor points:
* Notation in Eq 5. should be improved for better understanding
* What is the reason for the checkerboard artifacts shown in the progressive generation (Figure 3)?
* While theoretically each pixel only depends on log(L) latent variables

Update: The authors response has addressed my major concerns. Particularly, the revised version is more clear about how this paper relates to prior work, more comparisons/metrics are added, and issues I had related to clarity were addressed. The revised version has significantly expanded the scope of experiments, which better demonstrate the advantage of hierarchical features for normalizing flows. I have update my rating to marginal accept

---

> ### Author Response · Authors · 2020-11-24
> **Response to Reviewer 4**
>
>
> Thanks for reviewing our paper and pointing out some questions!
>
> ---
>
> 1. "more discussion how this paper relates to Neural Network Renormalization Group (Li & Wang). more background/examples .."
>
> Sure, we have added another paragraph in "Related work" section to discuss the background of renormalization group, flow-based models, and exact holographic mapping. In addition, we added more experiments to illustrate difference between RG-Flow and other flow-based models. And please kindly refer to general comments (1) and (2).
>
> ---
>
> 2. "no evaluation of the image inpainting performance"
>
> Thanks for the comment and we have added the metrics of PSNR and Inception-PSNR for inpainted images in Appendix F, which are consistent with the intuitive results in Fig. 7.
>
> ---
>
> 3. "Constrained NVP seems to be too weak a baseline for the image inpainting task .. more reasonable comparisons?  .. log(L) vs L^2 improvement, timing or flop counts should be used to compare to Real NVP."
>
> We claim that the dimension of the optimization space is $O(\log L)$ in RG-Flow, instead of $L^2$. Within this constrained smaller space, the original image can be recovered. To demonstrate an apple-to-apple comparison, we think it is fair to compare constrained RG-Flow to constrained Real NVP.
>
> The $O(\log L)$ advantage is more of a theoretical one and showing such a space exists is interesting by itself. The advantage in asymptotic complexity will be significant when L is large, but because of our limited computation resources, we can only implement L up to 32. Also, RG-Flow frequently involves non-contiguous access of data when re-ordering the variables into RG blocks as in Fig. 2(b), which is not optimized in current ML frameworks.
>
> ---
>
> 4. "The paper provides limited quantitative results and comparisons. There is no discussion of the 3D chair experiment in the appendix."
>
> We provide BPD scores to quantitatively assess the quality and diversity of generated images in Appendix G, which are widely adopted for flow-based models.
>
> Based on your comment, we have added FID score as another metric. Note that FID score is not widely used to assess flow-based models, because flow-based models can directly give the probability density of each generated sample, which GAN cannot do. BPD may be a more fair metric on datasets of specific fields like human faces and synthetic datasets, and it avoids the bias from the knowledge encoded in Inception-v3. Also, in Appendix F, we have added the metrics of PSNR and Inception-PSNR to quantitatively assess the quality of inpainted images. In Appendix B where we discuss our newly added synthetic toy datasets, we include the metrics of BPD and FID as well.
>
> The 3D Chair dataset is used to show a limitation of RG-Flow, which is discussed in "Discussion and conclusion" section. If the dataset only contains a few high-level factors, such as the 3D Chair dataset, it is hard to find explainable high-level disentangled representations in our network. The settings for 3D Chair dataset is mostly the same as for other experiments. We include the hyperparameters and results in Appendix G.
>
> ---
>
> 5. "Notation in Eq 5. should be improved"
>
> Sure, we provide some improvement in the revision. Eq. 5 is provided only as a rigorous mathematical description of what is happening in Fig. 2, and you could also read Fig. 2 for an intuitive graphical understanding.
>
> ---
>
> 6. "What is the reason for the checkerboard artifacts shown in the progressive generation (Figure 4)?"
>
> The checkerboard artifacts are related to the checkerboard mask used in the bipartite structure in our local bijective maps, similar to those in Real NVP. If we want to further reduce the artifact, we may use random masks in each layer, and add loss terms between the progressively generated images and downscaled ground truth images.
>
> ---
>
> 7. "While theoretically each pixel only depends on log(L) latent variables..."
>
> We apologize that we can not understand this question based on its description, which seems incomplete. Based on our best guess, we would like to include the reply to the other reviewer here for better clarification.
>
> We claim that the dimension of the optimization space is $O(\log L)$ in RG-Flow, instead of $L^2$. Within this constrained smaller space, the original image can be recovered. To demonstrate an apple-to-apple comparison, we think it is fair to compare constrained RG-Flow to constrained Real NVP.
>
> The $O(\log L)$ advantage is more of a theoretical one and showing such a space exists is interesting by itself. The advantage in asymptotic complexity will be significant when L is large, but because of our limited computation resources, we can only implement L up to 32. Also, RG-Flow frequently involves non-contiguous access of data when re-ordering the variables into RG blocks as in Fig. 2(b), which is not optimized in current ML frameworks.

---

### Official Review · AnonReviewer1 · 2020-10-28
**Interesting idea, but needs more experiments**

**Rating:** 6
**Confidence:** 4

**Review:**

Summary:
The paper introduces an RG-Flow model -- a hierarchical flow model based on the idea of the renormalization group. The experiments suggest that this model is expressive and capable of learning the disentangled representations at different scales. Additionally, the authors show how this method can be adapted for image inpainting and theoretically analyze its complexity.

Pros:
- Novel and interesting idea.
- The proposed method allows for separating the features at different scales (at low, mid, and high levels) due to the hierarchical structure of the model.

Concerns:
- The paper would benefit from additional experiments and comparison to other methods. For example, you can experiment with the datasets specific for the analysis of disentangled representations, e.g., Shapes 3D, MPI 3D, etc.
- The paper lacks the quantitative assessment of the degree of disentanglement of learned representations. I suggest to refer to the paper [1] for the list of disentanglement metrics. While they were invented for VAE-based models, you can adapt it to your method.
Similarly, you could add the assessment of the inpainting quality apart from the visual evaluation in Figure 6.

Questions:
- Did you measure the quality of generated images, e.g., by computing FID or Inception score? While the samples for the CelebA dataset look good, the samples for CIFAR-10 (Fig.8) are not impressive.
-  I did not find any details on the experiments on the 3D Chair dataset apart from Table 1 in the Appendix. Was the setting the same as for CelebA and CIFAR-10 datasets? Is it possible to compare the results of RG-Flow to Real NVP for the 3D Chair dataset?

Comments:
- I suggest adding the paper [2] to the related work; this paper introduces Multiscale Entanglement Renormalization Ansatz (MERA), a structure similar to your network.
- Take a look at the paper [3], bridging the entanglement renormalization and wavelets.

UPD: I am satisfied with the authors' response, and therefore I increase the score.

References:

[1] Locatello, F., Bauer, S., Lucic, M., Raetsch, G., Gelly, S., Schölkopf, B., & Bachem, O. (2019, May). Challenging common assumptions in the unsupervised learning of disentangled representations. In international conference on machine learning (pp. 4114-4124).

[2] Vidal, G. (2008). Class of quantum many-body states that can be efficiently simulated. Physical review letters, 101(11), 110501.

[3] Evenbly, G., & White, S. R. (2016). Entanglement renormalization and wavelets. Physical review letters, 116(14), 140403.

---

> ### Author Response · Authors · 2020-11-24
> **Response  to Reviewer 1**
>
> Thanks for taking time reviewing our paper and pointing out some questions.
>
> ---
> 1. "The paper would benefit from additional experiments and comparison to other methods .. specific for the analysis of disentangled representations .."
>
> A major focus and contribution of this work is on disentangling representations at different scales or spatially separated, as renormalization group guarantees the separation of information at different scales: it pushes the large-scale information to the high level layer and leaves the detailed information in the low level layer.
>
> In the updated "Experiments" section and Appendix B, we have added our synthetic toy datasets MSDS1 and MSDS2 to directly illustrate that RG-Flow can naturally capture global and local features separately. In comparison, we find Real NVP fails to capture the global and local characteristics of the datasets. For the human face dataset, we also add receptive fields learned by Real NVP in Appendix C. It directly shows that the mid level and low level representations of human faces are missing.
>
> For spatially separated features, they are disentangled in a sense that changing one will not affect the other, because their representations have non-overlapping receptive fields. For example, in the human face dataset, the mid-level factor of eyes will only change the eyes and not affect the hair, since they do not overlap.
>
> We think the disentanglement of high level representations are important questions. The disentanglement-specific methods, such as beta-VAE, use independence of latent variable as a drive to encourage high level disentanglement. In the flow based models, the latent variables are already independent. So we impose sparse prior as heuristic method to break the symmetry of latent space. But we are aware that sparsity does not guarantee disentanglement of high level representations [1]. In addition, we feel that disentangling just high-level factors with overlapping receptive fields is an ill-defined problem without inductive bias, which is also the main conclusion of [2]. So we acknowledge disentanglement of high level (overlapping) representations is another topic, which may be subject to our future work.
>
> [1] Geoffrey Roeder, et al., On Linear Identifiability of Learned Representations. (2020)
>
> [2] Francesco Locatello, et al., Challenging Common Assumptions in the Unsupervised Learning of Disentangled Representations. (2019)
>
> ---
>
> 2. "Similarly, you could add the assessment of the inpainting quality apart from the visual evaluation in Figure 6."
>
> Thanks for the comment and we have added the metrics of PSNR and Inception-PSNR for inpainted images in Appendix F, which are consistent with the intuitive results in Fig. 7.
>
> ---
> 3. "Did you measure the quality of generated images, e.g., by computing FID or Inception score? .."
>
> The quality of the generated images can be measured by bit per dimension (BPD), which is reported in Appendix G. Unlike generative adversarial networks, BPD can be directly computed in flow-based generative models.
>
> Our network achieves comparable BPD performance to Real NVP with approximately the same number of trainable parameters, when trained on CelebA and CIFAR-10 datasets. The performance of RG-Flow depends largely on the family of local bijective maps. Since we use the bipartite bijective maps, which are also used in Real NVP, Real NVP is a natural choice for the comparison. We expect that the advantage of RG-Flow can be transferred to more advanced bijective maps, such as invertible convolutions and i-ResNet to achieve better performance.
>
> Based on your suggestion, we also add FID score as another metric in Appendix G. Please note that FID score is not widely adopted to assess flow-based models since flow-based models can directly give the probability density of each generated sample, which GAN cannot. BPD may be a more fair metric on datasets of specific fields like human faces and synthetic datasets, and it avoids the bias from the knowledge encoded in Inception-v3.
>
> ---
>
> 4. "..Was the setting the same as for CelebA and CIFAR-10 datasets?"
>
> We will include the implementation details and results in Appendix G of the revision. The settings for 3D Chair dataset is mostly the same as for other experiments. This experiment is used to show a limitation of RG-Flow, which is discussed in "Discussion and conclusion" section. If the dataset only contains a few high-level factors, such as the 3D Chair dataset, it is hard to find explainable high-level disentangled representations in our network.
>
> ---
> 5. "I suggest adding the paper [2] to the related work.."
>
> Sure, we have added these important references to "Related work" section. In addition, we have also added some discussion of renormalization group and flow-based modeling.

---

### Author Response · Authors · 2020-11-24
**General comments**

We thank all reviewers for their constructive feedbacks.

In light of the comments from the reviewers, we have made several significant changes in the revision:

1. We add another paragraph in "Related work" section to discuss the previous works using renormalization group (RG) in tensor networks and neural networks. The original NeuralRG is designed especially for physical systems for Monte Carlo simulations (Li & Wang, 2018) or study of latent space geometry (Hu et al., 2020). However, it cannot be directly applied to complicated natural images and there has not been results on natural image datasets. Through our trials, directly applying those structures to natural images leads to abysmal results. To apply the idea of RG to more complicated datasets, we implement more expressive local bijective maps involving deep residue blocks, swish activation, and weight normalization etc. Our implementation makes RG-Flow capable of generating natural images. We are the first to thoroughly discuss the multi-scale latent representations under the RG transformation. An additional contribution is that we define the receptive fields of flow-based models and generation/inference causal cones for better theoretical understanding of RG-Flow.

2. To better illustrate that RG-Flow can naturally separate representations at different scales, we propose two synthetic toy datasets with multi-scale features, named MSDS1 and MSDS2. In "Experiments" section and in Appendix B, we show that RG-Flow can separately capture high-level and low-level features, whereas Real NVP fails to do so. The dataset will be publicly available. Further, in Appendix C, We show the receptive fields of Real NVP trained on CelebA dataset as a comparison to those of RG-Flow. We show that RG-Flow can produce semantically meaningful representations at each scale, and the progressive growth of latent representations from low level to high level is theoretically guaranteed. In contrast, Real NVP cannot produce semantically meaningful mid-level or low-level representations.

3. We provide more quantitative evaluations. In Appendix F, to quantitatively assess the quality of inpainted images, we add the metrics of PSNR and Inception-PSNR, and they are consistent with the intuitive results in Fig. 7. In Appendix G, we add FID score as another quantitative assessment of the quality and diversity of generated images. We have also updated BPD scores from new training results. Note that FID score is not widely adopted to assess flow-based models, because flow-based models can directly give the probability density of each generated sample, which GAN cannot do. BPD may be a more fair metric on datasets of specific fields like human faces and synthetic datasets, and it avoids the bias from the knowledge encoded in Inception-v3. See https://paperswithcode.com/sota/image-generation-on-cifar-10 for the adoption of those metrics.

In the following, we address each reviewer's comments specifically.

---

### Decision · Program_Chairs · 2021-01-07
**Final Decision**

**Decision:**

Reject

**Comment:**

This paper proposes a hierarchical flow-based generative model to learn disentangled features at different levels of abstractions.  The key technical contribution is a combination of renormalization group and flow-based models. The reviewers do find the idea interesting. However, the merit of the work with respect to StyleGAN and StyleFlow has not been well established. AR3 made the following comment:  “Specially, compared with the style-based generator[1,2], …, I don’t find superiorities of the proposed method.” The authors responded to the comment briefly (but not convincingly) in their rebuttal. There is no mention of it in the revised paper. A proper account of the issue would require major revision to the paper.